# How to Increase Job Satisfaction and Performance? Start with Thriving: The Serial Mediation Effect of Psychological Capital and Burnout

**DOI:** 10.3390/ijerph19138067

**Published:** 2022-06-30

**Authors:** Norberth Okros, Delia Vîrgă

**Affiliations:** Department of Psychology, West University of Timisoara, 300223 Timișoara, Romania; norberth.okros@e-uvt.ro

**Keywords:** thriving at work, psychological capital, burnout, job satisfaction, job performance

## Abstract

Based on the Job Demands-Resources and Broaden and Build theories, this study examines the role of thriving at work and psychological capital on burnout and, ultimately, job satisfaction and performance one year later. We used structural equation modeling to test the model on a sample of 317 Romanian correctional officers in a two-wave study at T1 and one year later (T2). The results indicate that thriving at work at Time 1 is positively linked to psychological capital at Time 1, and negatively related to burnout at Time 2. Meanwhile, burnout at Time 2 is negatively related to job satisfaction and performance at Time 2. The mediating chain effect of psychological capital at Time 1 and burnout at Time 2 is significant. Thriving at work and psychological capital are essential factors contributing to a decrease in subsequent burnout and increased job satisfaction and performance. In a thriving environment, correctional officers are more resilient, confident, optimistic, and hopeful at work, generating lower burnout over time and increasing job satisfaction and performance. Supervisors need to be encouraged to create a thriving work environment to increase psychological capital, reduce burnout, and improve correctional officers’ satisfaction and performance.

## 1. Introduction

Thriving at work has earned its well-deserved place in positive organizational psychology [1], a concept that is becoming increasingly studied. Its essential role in understanding the human dimension has recently been recognized and investigated in organizational and personal contexts. Based on the socially embedded model, thriving at work is defined as “a psychological state in which individuals experience both a sense of vitality and a sense of learning at work” [1] (p. 538), i.e., a temporary internal property that is related to the work context [1,2]. Vitality describes the available energy to complete work, the feeling of aliveness, and positive emotion. Learning is a function of developing and applying skills and abilities at work [1]. Previous research shows that the benefits of thriving are associated with the organizational dimension and improvements in individual aspects. From an organizational perspective, thriving at work has been associated with higher task performance [3], lower turnover intentions, higher levels of organizational citizenship, and higher organizational commitment [4]. At the individual level, the benefits are higher levels of subjective health, higher job satisfaction [2], and positive attitudes toward self-development [3].

Despite the increasing evidence focusing on the relationship between thriving at work and various desired individual resources and organizational outcomes, some gaps and divergences remain in the thriving literature. The present study addresses some of these.

First, most research on thriving at work relied on cross-sectional designs. As Kleine et al. [5] noted in their meta-analysis, only three studies assessed thriving at two measurement points, so we address this gap using a two-wave design with two measurement moments at a one-year time lag to study the role of thriving at work and personal resources in protecting against burnout and, ultimately, improving job satisfaction and performance. Second, in the literature, there is divergence regarding the relationship between thriving at work and PsyCap. Most research has shown that this positive state of mind (PsyCap) pushes employees to get involved in agentic work behaviors, thus promoting later thriving at work, e.g., [6]. Yet, this relationship can also be understood from another direction, as suggested by Shahid et al. [7]. Based on the results of this study, we make a significant contribution to this new perspective, namely, by studying the relationship between thriving and PsyCap. Third, we study a model with PsyCap and burnout as potential serial mediators in the relationship between thriving at work, job satisfaction, and performance. Hence, we also focus on the understudied relationship between personal resources and burnout, rarely investigated in two different moments. Last but not least, the beneficial role of thriving at work in increasing job satisfaction and performance has often been studied only in civil organizations, e.g., [4]. Another recommendation was that thriving at work be learned in organizations where the work environment is less civil [6]. In that context, this study highlights the impact of thriving at work on several variables within a sample of correctional officers.

In this study, we use Broaden and Build (B&B) theory of positive emotions [8] and the Job Demands-Resources (JD-R) theory [9] as theoretical frameworks. The B&B theory states that “certain discrete positive emotions—including joy, interest, contentment, pride, and love—although phenomenological distinct, all share the ability to broaden people’s momentary thought-action repertoires and build their enduring personal resources, ranging from physical and intellectual resources to social and psychological resources” [8] (p. 220). Through this theory, the positive emotions generated by thriving at work increase personal resources. More than ever, personal resources are helpful to employees in dealing with problematic situations, in general, and burnout, in particular, during the pandemic crisis, the JD-R theory argues that personal resources represent a protective factor in the face of occupational stress, referring to “the beliefs people hold regarding how much control they have over their environment” [9] (p. 275). This study used psychological capital (PsyCap) as a personal resource, a second-order concept that comprises four personal characteristics: self-efficacy, hope, optimism, and resilience [10].

Thus, based on the two theoretical frameworks, B&B and JD-R theories, we propose a model that assumes the existence of two sequential mediators of the relationship between thriving at work (T1) and the two outcomes: job satisfaction and performance (T2). This relationship is totally mediated by PsyCap (T1) and burnout (T2), and below we have detailed and argued each sequence from this model, formulating specific hypotheses.

### 1.1. Thriving at Work and Psychological Capital

The relationship between thriving at work and PsyCap is unclear in relation to their succession. In that order, recent studies show an association between PsyCap and thriving at work. For example, the first who demonstrated this relationship between PsyCap and thriving were Paterson et al. [3]; the relationship was later studied by Nawaz et al. [6]. Both studies showed that employees with a high level of PsyCap are more likely to be more energetic and learn more at work, meaning they can thrive. Yet, this relationship could also be explored from thriving to PsyCap, i.e., in the opposite direction [7]. In such a way, this relationship can be viewed from another perspective, namely the B&B theory of positive emotions [8]. Based on this theory, we argue that thriving at work ‘broadens’ thought-action repertoires because employees feel more energetic and can acquire and apply what they learn at work, which helps them to create new personal resources, such as PsyCap. We contend that vitality as a component of thriving will trigger various changes in thinking and, therefore, stimulate behaviors that will allow employees to develop new ideas, be more creative, and even help them create other types of resources. We also know that a high level of thriving at work allows individuals to be more resilient [11]. Learning at work facilitates the acquisition of new competencies and skills, leading to greater confidence and overcoming obstacles. Therefore, thriving at work will help employees and give them hope to achieve their goals in the future. Energetic correctional officers who learn at work likely enhance their positive attitude (optimism). When faced with different situations, they believe that they can overcome them easily (self-efficacy) and even solve challenging problems and recover quickly (resilient), having a motivational state that pushes them to achieve their goals (hope). Based on B&B theory, the relationship between thriving at work and PsyCap needs to be investigated in more detail [7]. Thus, in examining this relationship, we proposed Hypothesis 1:

**Hypothesis** **1** **(H1).**
*Thriving at work at T1 will be positively associated with PsyCap at T1.*


### 1.2. Psychological Capital and Burnout

PsyCap is defined as “an individual’s positive psychological state of development characterized by (1) having confidence (self-efficacy) to take on and put in the necessary effort to succeed at challenging tasks; (2) making a positive attribution (optimism) about succeeding now and in the future; (3) persevering toward goals and, when necessary, redirecting paths to goals (hope) to succeed; and (4) when beset by problems and adversity, sustaining and bouncing back and even beyond (resilience) to attain success” [10] (p. 3). Research has shown that PsyCap is an essential factor, positively related to performance and job satisfaction [12] and negatively related to stress and turnover [13], research has shown that high levels of PsyCap have been negatively related to burnout, e.g., [14,15], even in a Romanian sample, e.g., [16]. The JD-R theory states that personal resources are beneficial, protective factors that safeguard employees’ wellbeing and make them more resistant to burnout. Based on this theory, we argue that employees with a high level of PsyCap will experience lower levels of exhaustion and cynicism over one year based on the protective mechanism of personal resources. Research on the relationship between PsyCap and burnout at two different times is scarce. To make up for that, this research covers the gap and analyzes the relationship between personal resources and burnout using a two-wave design among correctional officers. So, we proposed hypothesis 2:

**Hypothesis** **2** **(H2).**
*PsyCap at T1 will be negatively associated with burnout at T2.*


### 1.3. Burnout and Job Satisfaction

Burnout is “a specific kind of occupational stress reaction among human service professionals, resulting from demanding and emotionally charged interactions with recipients” [17] (p. 424). In this study, we used core burnout (with its two components, emotional exhaustion and cynicism), an aspect often found in other research, e.g., [18,19]. Emotional exhaustion indicates feelings of depletion and being down, resulting from overtaxing work. Depersonalization or cynicism appears when individuals mentally distance themselves from their work by generating dehumanizing perceptions of co-workers, tasks, or clients [20]. Since burnout is problematic both for individuals and for the organization, it can be a factor that decreases employee satisfaction. As a job-related attitude, job satisfaction refers to “the extent to which people are satisfied with their work” [21] (p. 1). Job satisfaction is essential for an organization because it can bring many positive effects, such as organizational commitment and lower rates of turnover intention [22]. The JD-R theory demonstrates that job demands are mainly related to burnout, leading to adverse outcomes, such as lower job satisfaction. Previous empirical evidence supports the negative relationship between burnout and job satisfaction, e.g., [23]. In their study, Wang et al. [24] examined the relationship between job stress, burnout, job satisfaction, and organizational commitment in a sample of university teachers. They also divided teachers into three groups (national, provincial, and municipal), and multi-group results indicated a negative relationship between burnout and job satisfaction in all three cases. Plus, Cho et al. [25] demonstrated that correctional officers’ job demands led to emotional exhaustion and cynicism, reducing job satisfaction. In another study by Charoensukmongkol et al. [26], emotional exhaustion and depersonalization reduced employee satisfaction. As such, if employees experience high emotional exhaustion and cynicism, they will likely report lower job satisfaction. Consequently, we proposed hypothesis 3a:

**Hypothesis** **3a** **(H3a).**
*Burnout at T2 will be negatively associated with job satisfaction at T2.*


### 1.4. Burnout and Job Performance

Performance is among the essential outcomes of burnout. In-role performance is defined as “behaviors that are recognized by formal reward system and are part of the requirements as described in job descriptions” [27] (p. 606). The relationship between burnout and job performance is sustained and explained by the sixth proposition of the JD-R theory, which states that job strain negatively impacts job performance [9]. Employees with high burnout levels find it harder to cope with demands at work and perform poorer than employees with a low level of burnout [19]. Therefore, correctional officers with a high level of burnout do not have the energetic resources to reach their work goals. Studies have shown burnout’s negative effect on self-reported job performance [19,25,28] and objective performance indicators [29]. Taris’ [28] meta-analysis reported a negative correlation between exhaustion as part of core burnout and performance. In a study by Bakker and Heuven [17], the relationship between burnout and in-role performance was negative and significant among nurses and police officers. Consequently, we proposed hypothesis 3b:

**Hypothesis** **3b** **(H3b).**
*Burnout at T2 will be negatively associated with job performance at T2.*


### 1.5. Psychological Capital and Burnout as Serial Mediators

This study analyzes the potential serial mediation effect of PsyCap at T1 and burnout at T2 in the relationship between thriving at work at T1 and two outcomes, job satisfaction and performance at T2. This work contributes to the literature because PsyCap and burnout have not yet been studied as serial mediators, and not as separate ones using a two-wave design. Based on the B&B theory [8], we know that positive emotions are related to expanding thinking and action, thus building other personal resources. Plus, based on the JD-R theory [9], these personal resources are protective factors that help individuals increase their wellbeing. Thus, we argue that an energetic correctional officer who learns at work will report a higher level of positive emotions such as optimism, hope, self-efficacy, and resilience. Such an employee will experience less exhaustion and cynicism in the future (T2), contributing to job satisfaction and performance. Our focus is on PsyCap as a positive mediator and burnout as a negative mediator between thriving at work and outcomes (job satisfaction and performance). Boosted by thriving at work, PsyCap has a massive effect on reducing burnout [30]. Burnout is a debilitating state associated with low performance [19] and diminished job satisfaction [24]. Therefore, we suppose that correctional officers who thrive will face fewer challenges by maintaining a positive attitude and being more motivated to achieve their goals. They will have a greater capacity to recover from an unpleasant event, which will impact their job satisfaction and performance. Our fourth hypothesis is:

**Hypothesis** **4** **(H4).**
*The relationship between thriving at work at T1 and job satisfaction, and job performance at T2, is serially mediated by PsyCap at T1 and burnout at T2.*


## 2. Materials and Methods

### 2.1. Participants and Procedure

The sample consisted of correctional officers working in units administered by the National Administration of Penitentiaries (NAP), Romania. Data collection took place in two waves: in the fall of 2019 (T1), and one year later, in the fall of 2020 (T2), using a time-lagged study design. The sample was formed of correctional officers who worked in the operative sector of different prisons, meaning they had direct contact with the inmates. We used two waves of data collection (T1 and T2) to test the mediation path, based on Cole and Maxwell’s [31] recommendation. There was a one-year lag between T1 and T2, as suggested by the literature on PsyCap [32].

The nature of this type of organization with special procedures for data collection approval complicated the research implementation. All agreements to conduct this study were obtained from NAP (no. 12927/DMRU/15.01.2020) and the Ethics Committee of our University (no. 22848/0-1/26.05.2020), to be applied in six penitentiaries from the North and West parts of the country. Based on this agreement, we established together with each penitentiary unit how we could apply the correctional officers’ questionnaires. One psychologist from each unit distributed the questionnaires to correctional officers. The correctional officers participated voluntarily and in anonymity. No participant or correctional leadership individual received an incentive to participate in the survey in either T1 or T2.

In the first wave (T1), 400 questionnaires were distributed, and 350 were filled in and returned (87.5% response rate). Of the sample, 96.6% of respondents were men, and their ages ranged from 19 to 55 years (M = 35.41, SD = 8.82), with an average tenure of 12.61 years. Half of them had a higher educational background, were married, and had at least one child. In the second wave (T2), 350 questionnaires were distributed to the same participants who completed the questionnaires in the first wave. Among these participants, 317 filled in and returned the second questionnaire (90.57% response rate). This was a convenience sample, and its size reached the recommended limit for correlational stability [33]. In the next step, we performed t-tests to evaluate possible differences between respondents who participated in both waves and those who did not participate in wave 2. The results showed significant differences (*p* < 0.05) regarding age, marital status, and tenure. The 33 correctional officers who did not participate in the second wave had a significantly higher age (M = 43.09, SD = 3.32) and tenure (M = 22.86, SD = 5.72) than those who completed both waves. This could be explained by the possibility of those officers retiring by the second wave. Of those who saw it to completion, 96.8% of the complete sample were men aged 19 to 55 years (M = 34.30, SD = 8.47), with an average tenure of 11.50 years. One hundred and sixty-one (50.8%) had a higher educational background, 53.9% were married, and 45% had at least one child.

### 2.2. Measures

All variables were measured using Romanian versions of instruments that have been previously used (burnout – α = 0.81 [34]; job satisfaction − α = 0.90 [35]; performance – α = 0.75 [36]) or adapted (thriving at work – α = 0.86 [37]; PsyCap – α = 0.89 [38]) on Romanian samples. The instruments hold good psychometric properties, and we conducted confirmatory factor analyses on Romanian samples to ensure the internal consistency. Thriving at work and psychological capital were evaluated in the first wave (T1), and burnout, job satisfaction, and performance in the second wave (T2).

Thriving at work (T1) was measured with the adapted Thriving at Work Scale [2]. The scale includes four items for the learning dimension (e.g., “I see myself continually improving”) and five for vitality (e.g., “I have energy and spirit”). All items were rated using a seven-point Likert scale ranging from 1 (strongly disagree) to 7 (strongly agree).

PsyCap (T1) was measured with the Psychological Capital Questionnaire [12]. This questionnaire has four subscales: hope (“At the moment, I feel quite fulfilled at work”), self-efficacy (“I feel confident presenting information to a group of colleagues”), resilience (“Usually, at work, I easily get over the stressful aspects”), and optimism (“I am optimistic about what will happen to me in the future regarding my job”). All 12 items were scored on a six-point Likert-type scale (1 = strongly disagree, 6 = strongly agree).

Burnout (T2) was assessed with two scales of the Maslach Burnout Inventory-General Survey (MBI-GS; [39]): emotional exhaustion (five items; “I feel emotionally drained from my work”) and cynicism (four items; “I have become more cynical about whether my work contributes anything”). All items were scored on a seven-point scale ranging from 0 (never) to 6 (always).

Job satisfaction (T2) was measured with the Michigan Organizational Assessment Questionnaire [40]. The scale has three items with a response on a seven-point scale (1 = total disagreement, 7 = total agreement). A sample item reads: “In general, I like working here”.

Job performance (T2) was measured by a five-item questionnaire [27]. Participants self-rated their performance on a five-point scale (1 = strongly disagree, 5 = strongly agree). An example of an item is: “Adequately completes assigned duties”.

### 2.3. Data Analysis

The data were analyzed using structural equation modeling (SEM) from the *lavaan* package [41] in R software [42]. We chose to use the SEM technique because it evaluates both the measurement validity and complicated regression paths among the multiple variables [43], and, also, it has the advantage of analyzing multiple regressions. Moreover, we chose SEM because, first, we performed a series of confirmatory factor analyses (CFA) for some scales that we used (i.e., psychological capital, job performance, and satisfaction), the others having already been validated in previous studies (i.e., thriving at work and burnout). Second, we assessed four measurement models using CFA: M1—five factors model (thriving at work, PsyCap, burnout, job performance, and satisfaction), M2—four factors model (thriving at work, PsyCap, burnout, and the fourth latent factor consisting of job satisfaction and performance), M3—single-factor model, and M4—bifactor model (that includes two method factors, one for Time 1 items, and another for Time 2 items). Third, we tested four structural models: M5—hypothetical model, M6—PsyCap as an antecedent (in this model, PsyCap is passed as the antecedent of thriving, the rest identical to the hypothesized model), M7—partial mediation model (here, both PsyCap and burnout have been tested as partial mediators on the relationship between thriving at work, job satisfaction, and performance), and M8—serial model (in this model, we test serial mediation with thriving at work as a predictor variable, PsyCap and burnout as mediating variables, job satisfaction as an outcome variable, and job performance as a dependent variable after satisfaction).

We used the factor scores as indicators in the structural models. Thus, thriving at work was treated as a latent factor composed of vitality and learning at work. PsyCap was composed of its four manifest dimensions (self-efficacy, hope, optimism, and resilience), and burnout was composed of emotional exhaustion and cynicism. The model fit was evaluated using maximum likelihood estimation, and the indirect effects were evaluated using 5000 bootstrap samples with 95% confidence intervals. We calculated three absolute fit indices (chi-square statistic; root mean square error of approximation, RMSEA; standardized root mean square residual, SRMR) and two relative fit indices (Tucker Lewis index, TLI; comparative fit index, CFI). Cut-off criteria indicating a satisfactory model fit were: RMSEA < 0.06, CFI and TLI > 0.95, and SRMR < 0.08 [44]. We also used a chi-square difference test (∆χ2; [45]) to assess the model fit.

Before moving on to the results section, several limitations should be mentioned, such as the fact that we used thriving at work as a latent variable composite, how we only used self-reported measurements, or the lack of panel data needed to make a cross-lagged analysis between variables measured in two different moments. All these limitations are explained and developed in “Section 4.2. Limitations and directions for future research”.

## 3. Results

Means, standard deviations, Cronbach’s alpha values, and correlations among the study’s variables are presented in Table 1. Almost all correlations between variables were statistically significant, and all reliabilities were acceptable, from 0.72 for resilience to 0.93 for job satisfaction.

We conducted a confirmatory factor analysis (CFA) before testing the hypotheses, and the CFA for measurement models revealed that the five factors model (M1) had a better fit compared to the four factors model (M2), the single-factor model (M3), and bi-factor model (M4). The results show that the five-factor model (M1: χ2(94) = 143.181, *p* < 0.001, TLI = 0.98, CFI = 0.98, RMSEA = 0.04, SRMR = 0.03) was superior to the four-factors model (M2), single-factor model (M3), and bi-factor model (M4) Therefore, there is a low chance that common method bias appeared (see Table 2). Further, we compared the three structural models, obtaining acceptable fit indices for all three. However, in the model with PsyCap as an antecedent (M6), the relationship between thriving at work at T1 and burnout at T2 was non-significant (β = −0.07, *p* > 0.05). This was also the case in the partial mediation model (M7), in which neither the relationship between thriving at work at T1 and burnout at T2 (β = 0.10, *p* > 0.05), nor the one between PsyCap at T1 and job satisfaction at T2, were significant (β = −0.12, *p* > 0.05). Last but not least, in the case of the serial model (M8), the mediation effect was statistically non-significant. Therefore, based on the theoretical framework of this research, the acceptable model was the hypothetical one (M5). More precisely, the three models (M5, M6, and M7) confirmed that thriving at work is positively associated with personal resources (PsyCap) that can be used in difficult or challenging times.

As depicted in Figure 1, thriving at work at T1 was positively related to PsyCap at T1 (β = 0.60, *p* < 0.001), PsyCap at T1 was negatively related to burnout at T2 (β = −0.17, *p* < 0.01), and burnout at T2 was negatively related to job satisfaction at T2 (β = −0.69, *p* < 0.001) and job performance at T2 (β = −0.47, *p* < 0.001). Thus, all hypotheses were supported by data.

More importantly, the serial mediating effects (see Table 3) of PsyCap at T1 and burnout at T2 were significant both for job satisfaction (β = 0.05, *p* < 0.05, 95% CI [0.01, 0.10]) and performance (β = 0.07, *p* < 0.05, 95% CI [0.02, 0.27]). Our data supported the hypothesized model, and the standardized estimates for each relationship are presented in Figure 1. Plus, the hypothesized model explained considerable variance in the mediators, PsyCap (R2 = 0.36), burnout (R2 = 0.03), and the two outcomes, job satisfaction (R2 = 0.48) and job performance (R2 = 0.22).

## 4. Discussion

This research investigated the role of thriving at work and PsyCap, as a personal resource, on later consequences (burnout, job satisfaction, and performance one year later). Based on the B&B and JD-R theories, we tested these relationships among Romanian correctional officers using time-lagged data. The results support all four hypotheses of the study and the model.

First, according to the B&B theory, positivity produced by thriving at work broadens people’s building of additional resources, such as PsyCap. Thus, we demonstrated that thriving at work is positively associated with PsyCap. Personal resources (such as PsyCap) are necessary for a work environment like that of a correctional officer. Therefore, an energetic correctional officer, who learns new strategies at work, tends to be more hopeful in achieving work goals, with a positive attitude and resilience to overcome any obstacle. Second, we showed the role of PsyCap, as a personal resource, in employees’ wellbeing, reducing their emotional exhaustion and depersonalization over a one-year time lag. We analyzed this relationship based on the JD-R theory, and while the results were consistent with previous literature, e.g., [15,16], these also provided additional evidence from a time-lagged perspective. Third, we analyzed the relationship between burnout and job satisfaction on one side, and performance on the other. We expected burnout to be very problematic for both individuals and the organization. In line with this expectation, our results showed that high levels of emotional exhaustion and depersonalization make correctional officers less satisfied with their work and less performant in their job. These results align with previous empirical evidence, e.g., [24,25].

Last but not least, our model focused on the serially mediated relationship between thriving at work and the two outcomes (satisfaction and performance) via PsyCap and burnout. The results indicated that correctional officers stimulated by thriving at work have a higher level of PsyCap that reduces burnout over time. In turn, the burnout of correctional officers affects their satisfaction and performance after one year. Therefore, thriving correctional officers who are energetic and learn at work will face challenges and crises, have a positive attitude, and be motivated to achieve their goals. They will have a greater ability to recover from unpleasant events, and these things impact job satisfaction and performance one year later. Thus, thriving has a potentiating role for PsyCap in this environment, and also, PsyCap acts as a protector for thriving employees who report less burnout over time. In this vein, the relationship above predicts diminished satisfaction with their work and performance as correctional officers. These results expand the negative association between PsyCap and burnout, previously identified in cross-sectional studies and samples of nurses or programmers [16,30].

Thriving at work as the employed predictor and the explanatory mediating mechanisms (i.e., PsyCap and burnout) can explain almost half of the job satisfaction variance and one-quarter of performance variance. This indicates an efficient model that accurately identifies correctional officers’ job satisfaction and performance triggers.

### 4.1. Theoretical and Practical Implications

Based on these results, the present research has several theoretical and practical contributions, covering several gaps. The first contribution of this research is that we provide evidence on the assumption of Shahid et al. [7], related to the relationship between thriving at work and PsyCap, in that order, based on B&B theory. The results revealed a positive link between thriving at work and PsyCap, which shows that the positive emotions generated by thriving at work lead to increased psychological capital. Plus, another contribution shows that this positive link can be a starting point for future studies to test, in longitudinal designs, whether thriving at work is an antecedent for PsyCap. Furthermore, we responded to need highlighted by Porath et al. [2] for thriving to be analyzed with other positive constructs (such as PsyCap).

Another contribution of this study is investigating the relationship between PsyCap and burnout using time-lagged data. Therefore, we covered a gap in the literature regarding investigating these effects over time, as mentioned by Kleine and colleagues [5]. We have shown that PsyCap can reduce burnout experiences even at a one-year distance. These results specifically contribute to understanding the role of PsyCap in JD-R theory. Previous research showed that PsyCap helps reduce burnout, e.g., [14,16]. Therefore, these results provide evidence for the JD-R theory by showing that PsyCap is a personal resource with a protective role in triggering burnout over time.

PsyCap and burnout acted as serial mediators of the relationship between thriving and job satisfaction and performance. This is the first study that fits together the role of the two concepts in such a mediation model. Thus, our results contribute to the B&B and the JD-R theories, showing that positive emotions lead to the building of other personal resources, which, in turn, are protective factors for individuals in relation to stress, and also lead to an increase in wellbeing in time. Beyond that, this study brings evidence for the JD-R theory about personal resources’ role in burnout from a longitudinal perspective.

Last but not least, the beneficial role of thriving at work has so far been demonstrated only in civil organizations, e.g., [4]. Through this study, we show the valuable role of thriving even in environments characterized by a hierarchical institutional culture, in which the rules are stringent, with various and frequent stressful situations, some of which even endanger the lives of correctional officers from Romania.

This study also has some practical implications. Based on previous research, e.g., [11], we know that various organizational strategies, such as decision-making discretion, minimizing incivility, or providing performance, cultivate thriving at work, which, in turn, promotes the development of individuals and also contributes to positive health. Therefore, the NAP should create an environment to encourage superiors to provide feedback on employees’ performance, reduce incivility, and thus increase thriving among correctional officers.

The results also showed that thriving is related to PsyCap. As a personal resource, it is malleable and can be improved through specific training and interventions, helping correctional officers better adapt to working conditions. According to a meta-analysis by Lupșa et al. [46], this training can last from one hour to four weeks and consists of exercises to increase employees’ psychological resources. These resources protect employees against burnout, which, as we have seen, affects job satisfaction and performance.

As practical implications, Romanian correctional officers who experiment thriving at work (i.e., feel energetic at work and accumulate knowledge in the workplace) will develop positive emotions related to their confidence that they can complete a task, they have an attitude that positively evaluates everything around them, or they have a belief that they will achieve their goals and that they can recover after facing a problem. The correctional officer characterized by these positive states will experience less exhaustion and cynicism in the future, which will make him more satisfied at work and report a higher performance. Thus, a climate of thriving matters for correctional officers because it enhances employees’ health and positive adaptation to work stressors, creating a healthier organizational environment.

### 4.2. Limitations and Directions for Future Research

Several limitations should be considered when interpreting the current findings. First, in the present research, we have used thriving at work as a composite latent variable, including both dimensions, as was done in previous research, e.g., [2]. Future research should also analyze the two dimensions separately, concerning other variables. Second, we used only self-reported measurements in this study. Future research should employ objective data, such as supervisor ratings regarding performance. Third, the lack of panel data restricted us from making other types of analysis (like examining the reversal relationship). Penitentiaries could be considered hard-to-reach participants because of their geographical locations in different parts of the country. Researchers need specific approval and have limited time to administer the survey at each site. Plus, the restrictions imposed by pandemic time added new limits to access on participants, and we needed to measure different variables in T1 (thriving and PsyCap) and T2 (burnout, job satisfaction, and performance). Using the cross-lagged design, future research needs panel data to analyze the reversal relationships between these helpful protective factors and positive outcomes. Fourth, although this study examined two serially mediating mechanisms, PsyCap and burnout, future research should seek to build a complete picture of how thriving at work can impact job satisfaction and performance via PsyCap and burnout. Future research could also explore other mechanisms (e.g., psychological needs satisfaction), contributing to improving our understanding of this relationship using multilevel designs.

## 5. Conclusions

In conclusion, this time-lagged study reveals the serially mediating effects of PsyCap and burnout on the relationship between thriving at work, job satisfaction, and performance, focusing on correctional officers. We also showed that thriving at work is related to PsyCap, which affected job satisfaction and performance through burnout among Romanian correctional officers. PsyCap is crucial to reducing officers’ burnout in time because officers with a higher level of PsyCap are less likely to become emotionally exhausted and cynical about their work. Therefore, this study can be a starting point for creating various practices and interventions that can be used to increase personal resources to protect employees from burnout and improve organizational satisfaction and performance.

## Figures and Tables

**Figure 1 ijerph-19-08067-f001:**
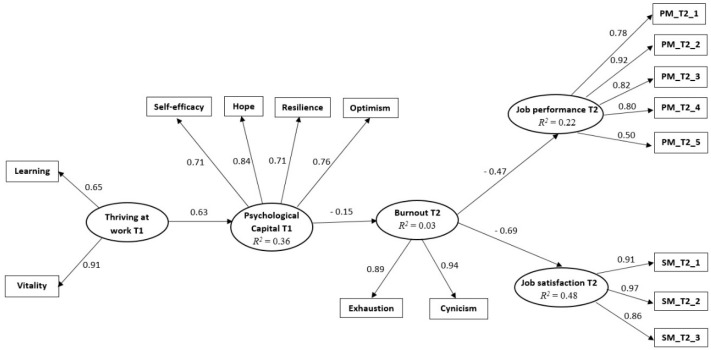
The results for the hypothesized model. Note: The standardized coefficients are illustrated.

**Table 1 ijerph-19-08067-t001:** Means, standard deviations, and correlation coefficients between variables.

Variable	*M*	*SD*	1.	2.	3.	4.	5.	6.	7.	8.	9.	10.
1. Learning T1	5.86	0.83	(0.79)									
2. Vitality T1	5.75	0.88	0.59 **	(0.77)								
3. Self-efficacy T1	4.82	0.95	0.29 **	0.38 **	(0.90)							
4. Hope T1	4.84	0.71	0.36 **	0.49 **	0.61 **	(0.73)						
5. Resilience T1	4.88	0.76	0.19 **	0.36 **	0.52 **	0.60 **	(0.72)					
6. Optimism T1	5.01	0.79	0.37 **	0.49 **	0.53 **	0.63 **	0.54 **	(0.78)				
7. Exhaustion T2	4.67	5.25	0.01	−0.06	−0.14 *	−0.17 **	−0.17 **	−0.09	(0.92)			
8. Cynicism T2	2.74	3.94	0.03	−0.04	−0.12 *	−0.11 *	−0.11 *	−0.06	0.83 **	(0.93)		
9. Job satisfaction T2	6.13	1.04	0.02	0.12 *	0.07	0.09	0.13*	0.03	−0.61 **	−0.61 **	(0.93)	
10. Performance T2	4.23	0.50	0.02	0.03	0.01	−0.01	−0.04	−0.05	−0.40 **	−0.42 **	0.48 **	(0.85)

Notes. *N* = 317, * *p* < 0.05; ** *p* < 0.01; Cronbach’s α coefficients are displayed on the main diagonal.

**Table 2 ijerph-19-08067-t002:** Fit indices and model comparisons for measurement and structural models.

Model	χ2	*df*	χ2/*df*	CFI	TLI	RMSEA [90% CI]	SRMR	∆χ2	∆*df*
Measurement model									
M1-five factors model	143.181 ***	94	1.52	0.98	0.98	0.04 [0.03, 0.05]	0.03		
M2-four factors model	801.753 ***	98	8.18	0.78	0.73	0.15 [0.14, 0.16]	0.11	659.572	4
M3-single-factor model	1761.806 ***	104	16.94	0.48	0.40	0.22 [0.21, 0.23]	0.19	1618.625	6
M4-bi-factor model	1957.519 ***	616	3.17	0.84	0.82	0.08 [0.08, 0.09]	0.06	1814.338	522
Structural model									
M5-hypothetical model	122.989 *	97	1.26	0.99	0.99	0.03 [0.01, 0.04]	0.04		
M6-PsyCap as antecedent	160.533 ***	99	1.62	0.98	0.97	0.04 [0.03, 0.06]	0.04	37.544	2
M7-partial mediation model	143.181 **	94	1.52	0.98	0.98	0.04 [0.03, 0.05]	0.03	20.192	3
M8-serial model	161.613 ***	100	1.61	0.98	0.97	0.04 [0.03, 0.06]	0.04	38.624	3

Note. *N* = 317. For the M2, M3, and M4 model, the comparison is versus M1. M6, M7 and M8 is compared to M5; * *p* < 0.05, ** *p* < 0.01, *** *p* < 0.001

**Table 3 ijerph-19-08067-t003:** Standardized indirect effects with bootstrapped 95% confidence intervals.

Independent Variable	Mediator 1	Mediator 2	Dependent Variable	Estimate	95% CI
Thriving at work		PsyCap		Burnout		Job satisfaction	0.05 *	[0.01–0.10]
	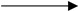		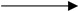		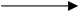			
Thriving at work		PsyCap		Burnout		Performance	0.07 *	[0.02–0.27]
	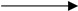		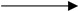		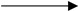			

Note. *n* = 317, * *p* < 0.05; Cronbach’s α coefficients are displayed on the main diagonal.

## Data Availability

The data presented in this study are openly available in [OSF] at [doi:10.17605/OSF.IO/9Y8KX].

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
