# Peer review of "How to Increase Job Satisfaction and Performance? Start with Thriving: The Serial Mediation Effect of Psychological Capital and Burnout"

_ijerph, 2022, doi:10.3390/ijerph19138067_

Round 1

Reviewer 1 Report

Thank you for opportunity to review your interesting and informative article – it is well developed, well written and merits publication. I do, however, recommend several revisions. First, as author(s) you need to incorporate divergent and conflicting perspectives which are evident in the literature (2-3 paragraphs). Secondly, from a methodology perspective, I recommend you shift the limitations from the back end of the paper to Materials and Methods section so as to enable the reader audience to understand the limitations and any caveats which impact or have an influence related to the research the opportunity to place the paper into a context in relation to what is stated / proposed by you. As well, you need to identify the rationale as to how and why the country was selected, and then how the specific in country location(s) were determined / selected; if the survey instrument were pre-tested and under what conditions and to identify the number of unusable surveys in comparison to returned surveys for both T1 and T 2. As well, did any participant or correctional leadership individual receive an incentive to participate in the survey in either T 1 / T2.

I hope you will take the time to incorporate the suggestions revisions to take your paper to the next step – publication; since doing so, will also ensure you gain a reader following for your interesting and informative research.

Author Response

Manuscript ID ijerph-1792899

Title How do increase job satisfaction and performance? Start with thriving: the serial mediation effect of psychological capital and burnout

Reviewer 1

Thank you for opportunity to review your interesting and informative article – it is well developed, well written and merits publication. I do, however, recommend several revisions. First, as author(s) you need to incorporate divergent and conflicting perspectives which are evident in the literature (2-3 paragraphs).

 Secondly, from a methodology perspective, I recommend you shift the limitations from the back end of the paper to Materials and Methods section so as to enable the reader audience to understand the limitations and any caveats which impact or have an influence related to the research the opportunity to place the paper into a context in relation to what is stated / proposed by you. As well, you need to identify the rationale as to how and why the country was selected, and then how the specific in country location(s) were determined / selected; if the survey instrument were pre-tested and under what conditions and to identify the number of unusable surveys in comparison to returned surveys for both T1 and T2. As well, did any participant or correctional leadership individual receive an incentive to participate in the survey in either T1 / T2.

RESPONSE: Thank you for your valuable feedback and suggestions throughout this review process! We have appreciated your guidance and did our best to integrate them with comments from the other reviewers and the study's objectives. We reply to the comments below.

Thus, we have developed and added to the Introduction section more information about gaps and divergences in the literature. Here, we have described the contribution and possible clarifications we bring to the literature on thriving at work through this research. We have referred to the fact that (1) using a two-wave design with two measurement moments at a one-year time lag to study the role of thriving at work and personal resources against burnout and, ultimately, to job satisfaction and performance, to the fact that (2) we have brought a clarification regarding the understanding from another direction of the relationship between thriving and PsyCap, to the fact that (3) we have analyzed the relationship between personal resources and burnout, rarely investigated in two different moments, and last but not least, to the fact that (4) the role of thriving at work did not have been studied in organizations where the work environment is less civil.

We have mentioned all these things on pp. 1-2:Despite the increasing evidence focusing on the relationship between thriving at work and various desired individual resources and organizational outcomes, some gaps and divergences remain in the thriving literature. The present study addresses some of these.

First, most research on thriving at work relies on cross-sectional designs. As Kleine et al. [5] note in their meta-analysis, only three studies assessed thriving at two measurement points, so we address this gap using a two-wave design with two measurement moments at a one-year time lag to study the role of thriving at work and personal resources against burnout and, ultimately, to job satisfaction and performance. Second, in the literature, there is divergence regarding the study of the relationship between thriving at work and PsyCap. Most research has shown that this positive state of mental development (PsyCap) pushes employees to get involved in agentic work behaviors, thus promoting later thriving at work [e.g., 6]. But this relationship can also be understood from another direction suggested by Shahid et al. [7]. Thus, in this study, we make a significant contribution that supports this new perspective: the study of the relationship from thriving to PsyCap. Third, we studied a model with PsyCap and burnout as potential serial mediators in the relationship between thriving at work, job satisfaction, and performance. Hence, we also focus on the understudied relationship between personal resources and burnout, rarely investigated in two different moments. Last but not least, the beneficial role of thriving at work indirectly increases job satisfaction and performance has often been studied only in civil organizations [e.g., 4]. Another recommendation was that thriving at work be learned in organizations where the work environment is less civil [6]. Therefore, this study highlights the impact of thriving at work on several variables within a sample of correctional officers.”

R1: Secondly, from a methodology perspective, I recommend you shift the limitations from the back end of the paper to the Materials and Methods section so as to enable the reader audience to understand the limitations and any caveats which impact or have an influence related to the research the opportunity to place the paper into a context in relation to what is stated/proposed by you.

RESPONSE: Thank you for your suggestion! From several points of view, we think it is not very useful to move the whole section 2.4. Limitations and directions for future research within methods and materials. First, the structure recommended by the APA is very precise in terms of the structure of a research article. Second, the limits section refers to the study's limits after the results are known to provide suggestions for future research. In the Methods and Materials section, the results are not yet known, and if we added the section here, this would create misunderstandings for readers. However, to address your suggestion, we have added a short paragraph at the end of the Methods and Material section (p. 6), in which we listed the limitations, mentioning that they are developed at the end: “Before moving on to the results section, several limitations should be mentioned, such as the fact that we have used thriving at work as a latent variable composite, we used only self-reported measurements, or the lack of panel data needed to make a cross-lagged analysis between variables measured in two different moments. All these limitations are explained and developed in section 2.4. Limitations and directions for future research.”

R1: As well, you need to identify the rationale as to how and why the country was selected, and then how the specific in country location(s) were determined / selected; if the survey instrument were pre-tested and under what conditions and to identify the number of unusable surveys in comparison to returned surveys for both T1 and T2. As well, did any participant or correctional leadership individual receive an incentive to participate in the survey in either T1 / T2.

RESPONSE: Thank you for your recommendation! Indeed, any participant or correctional leadership individual did not receive an incentive to participate in the survey in either T1 or T2. We have added these sentences to the Participants and Procedure section: “This was a convenience sample, and its size reached the recommended limit for correlational stability [33].” andAny participant or correctional leadership individual did not receive an incentive to participate in the survey in either T1 or T2”.

 Regarding the aspects related to the country in which we conducted the study, in addition to the explanations in the text, it should be mentioned that very few studies address issues related to Romanian prisons because they need a lot of approval and because they are quite skeptical. Regarding the aspects related to another country, it would have been even more challenging for us, as foreigners, to be able to carry out such research. Last but not least, we chose those six penitentiaries because we had previous collaborations with psychologists working in them, and we benefited from their help, particularly on the second survey.

Regarding the questionnaires, all were used or adapted to the Romanian population, this being mentioned in the article (p. 5): “All variables were measured using Romanian versions of instruments that have been previously used (burnout – α = .81, [34]; job satisfaction – α = .90, [35]; performance – α = .75; [36]) or adapted (thriving at work – α = .86; [37]; PsyCap – α = .89; [38]) on Romanian samples.” Moreover, the thriving at work scale was validated on the Romanian population, using even a sample of correctional officers.

Finally, on page 5, you will also find information on how many participants we lost from T1 to T2: “In the first wave (T1), 400 questionnaires were distributed, and 350 were filled in and returned (87.5% response rate). 96.6% of respondents were men, and their ages ranged from 19 to 55 years (M = 35.41, SD = 8.82), with an average tenure of 12.61 years. Half of them had a higher educational background, were married, and had at least one child. In the second wave (T2), 350 questionnaires were distributed to the same participants who completed the questionnaires in the first wave. Among these participants, 317 filled in and returned the second questionnaire (90.57% response rate). This was a convenience sample, and its size reached the recommended limit for correlational stability [33]. In the next step, we performed t-tests to evaluate possible differences between respondents who participated in both waves and those who did not participate in wave 2. The results showed significant differences (p < .05) regarding age, marital status, and tenure. The 33 correctional officers who did not participate in the second wave had significantly higher age (M = 43.09, SD = 3.32) and tenure (M = 22.86, SD = 5.72) than those who completed both waves. This could be explained by the possibility of those officers retiring until the second wave.”

R1: I hope you will take the time to incorporate the suggestions revisions to take your paper to the next step – publication; since doing so, will also ensure you gain a reader following for your interesting and informative research.

RESPONSE: Thank you very much once again for the feedback and suggestions provided! We hope that our additions are clear and welcome.

Reviewer 2 Report

Dear authors and esteemed editor,

Thank you for giving me the opportunity to review the submitted manuscript for possible publication in IJERPH. The article presents an interesting proposal in which thriving at work has a serial impact on psychological capital, job satisfaction, and burnout.

The study uses data from a sample of 317 correctional officers working in institutions in Romania. Broadly speaking, we can say that the research is part of the positive organizational psychology research tradition, showing that promoting thriving at work is a mechanism for making organizations healthier.

Research shows that thriving at work produces positive effects (T1) on psychological capital, and also identifies serial effects of variable magnitude on job satisfaction and burnout.

I consider that the article may be of interest to the audience of the journal due to the findings found and the type of methodology applied (serial SEM models applied to two waves).

However, I make several suggestions to improve the overall quality of the work:

- In the first place, the theoretical introduction is too concise and should show greater depth when explaining the theoretical connection between the different constructs evaluated, in particular between thriving at work and psychological capital. This without neglecting to explain the connection between thriving at work and satisfaction and burnout.

- Secondly, I advise justifying (and citing a supporting reference) the selection of SEM models instead of other types of techniques whose results may be easier to interpret.

- I recommend extending the section in which both the theoretical contributions of the study and the practical implications are indicated. In the latter case, indicating how promoting work contexts characterized by thriving can turn organizations into healthier environments.

- Finally, it would be advisable to explain how, in the profession in which the sample works, and in the organizational context in which they work, the results of this article can serve to improve the quality of work life, reduce burnout and increase organizational performance.

I congratulate the authors for their work.

Warm regards,

Author Response

Manuscript ID ijerph-1792899

Title How do increase job satisfaction and performance? Start with thriving: the serial mediation effect of psychological capital and burnout

Reviewer 2

Dear authors and esteemed editor,

Thank you for giving me the opportunity to review the submitted manuscript for possible publication in IJERPH. The article presents an interesting proposal in which thriving at work has a serial impact on psychological capital, job satisfaction, and burnout.

The study uses data from a sample of 317 correctional officers working in institutions in Romania. Broadly speaking, we can say that the research is part of the positive organizational psychology research tradition, showing that promoting thriving at work is a mechanism for making organizations healthier.

Research shows that thriving at work produces positive effects (T1) on psychological capital, and also identifies serial effects of variable magnitude on job satisfaction and burnout.

I consider that the article may be of interest to the audience of the journal due to the findings found and the type of methodology applied (serial SEM models applied to two waves).

RESPONSE: Thank you for your valuable feedback and suggestions throughout this review process! We have appreciated your guidance and did our best to integrate them with comments from the other reviewers and the study's objectives. We reply to all comments below.

R2: However, I make several suggestions to improve the overall quality of the work:

In the first place, the theoretical introduction is too concise and should show greater depth when explaining the theoretical connection between the different constructs evaluated, in particular between thriving at work and psychological capital. This is without neglecting to explain the connection between thriving at work and satisfaction and burnout.

RESPONSE: First, in the theoretical introduction, we have added a new paragraph in which we have clarified our hypothetical model to make it easier to understand how we constructed and argued the hypotheses (p. 2):Thus, based on the two theoretical frameworks, B&B and JD-R theories, we propose a model that assumes the existence of two sequential mediators on the relationship between thriving at work (T1) and the two outcomes: job satisfaction and performance (T2). This relationship is totally mediated by PsyCap (T1) and burnout (T2), and below, we have detailed and argued each sequence from this model, formulating specific hypotheses.

We have also developed the argumentation for the hypothesis between thriving and PsyCap (pp. 2-3): The relationship between thriving at work and PsyCap is unclear related to the succession. In that order, recent studies show an association between PsyCap and thriving at work. For example, the first who demonstrated this relationship between PsyCap and thriving are Paterson et al. [3]; the relationship was later studied by Nawaz et al. [6]. Both studies show that employees with a high level of PsyCap are more likely to be more energetic and learn more at work, meaning they can thrive. But this relation could be explored from thriving to PsyCap direction [7]. Thus, this relationship can be viewed from another perspective, namely the B&B theory of positive emotions [8]. Based on this theory, we argue that thriving at work ‘broaden’ thought-action repertoires because employees feel more energetic and can acquire and apply what they learn at work, which helps them to create new personal resources, such as PsyCap. We contend that vitality as a component of thriving will trigger various changes in thinking and, therefore, stimulate behaviors that will allow employees to develop new ideas, be more creative, and even help them create other types of resources. We also know that a high level of thriving at work allows individuals to be more resilient [12]. Learning at work facilitates the acquisition of new competencies and skills, leading to greater confidence and overcoming obstacles. Therefore, thriving at work will help employees and give them hope to achieve their goals in the future. Energetic correctional officers who learn at work likely enhance positive attitude (optimism). When faced with different situations, they believe that they can overcome them easily (self-efficacy) and even solve challenging problems and recover quickly (resilient), having a motivational state that pushes them to achieve their goals (hope). Based on B&B theory, the relationship between thriving at work and PsyCap needs to be investigated in more detail [7]. Thus, examining this relationship, we proposed Hypothesis 1:”

R2:  Secondly, I advise justifying (and citing a supporting reference) the selection of SEM models instead of other types of techniques whose results may be easier to interpret.

RESPONSE: Regarding the justification for choosing SEM models, we have added in the Data analysis section a paragraph by which we motivate the choice made and mention what are the benefits of this type of analysis (p. 6): “The data were analyzed using structural equation modeling (SEM) from the lavaan package [40] in R software [41]. We chose to use the SEM technique because it evaluates both the measurement validity and complicated regression paths among the multiple variables [42] and also has the advantage of analyzing multiple regressions. Moreover, we chose SEM because…”

R2:  I recommend extending the section in which both the theoretical contributions of the study and the practical implications are indicated. In the latter case, indicating how promoting work contexts characterized by thriving can turn organizations into healthier environments.

RESPONSE: Also, we have developed and added other aspects regarding the Theoretical and Practical contributions. We hope this section is now more structured and explicit (pp. 9-10):

Based on these results, the present research has several theoretical and practical contributions, covering several gaps. The first contribution of this research is that we provide evidence on the assumption of Shahid et al. [7], related to the relationship between thriving at work and PsyCap, in that order, based on B&B theory. The results revealed a positive link between thriving at work and PsyCap, which shows that the positive emotions generated by thriving at work lead to increased psychological capital. Also, another contribution indicates that this positive link can be a starting point for future studies to test in longitudinal design whether thriving at work is an antecedent for PsyCap. Furthermore, we respond to needs highlighted by Porath et al. [2] that thriving be analyzed with other positive constructs (such as PsyCap).

Another contribution of this study is investigating the relationship between PsyCap and burnout using time-lagged data. Therefore, we cover a gap in the literature regarding investigating these effects in the time mentioned by Kleine and colleagues [5]. We have shown that PsyCap can reduce burnout experiences even at a one-year distance. These results specifically contribute to understanding the role of PsyCap in JD-R theory. Previous research has shown that PsyCap helps reduce burnout [e.g., 14, 16]. Therefore, these results provide evidence for the JD-R theory by offering that PsyCap is a personal resource with a protective role in triggering burnout over time.

PsyCap and burnout acted as serial mediators in the relationship between thriving and job satisfaction and performance. This is the first study that fits the role of the two concepts in such a mediation model. Thus, our results contribute to the B&B and the JD-R theories, showing that positive emotions lead to the building of personal resources, which in turn are protective factors for individuals in relation to stress, and also lead to an increase in well-being in time. Also, this study brings evidence for the JD-R theory about personal resources' role in burnout from a longitudinal perspective.

Last but not least, the beneficial role of thriving at work has been demonstrated only in civil organizations so far [e.g., 4]. Through this study, we could show the valuable role of thriving even in environments characterized by a hierarchical institutional culture, in which the rules are stringent, with various and frequent stressful situations, some of which even endanger the lives of correctional officers from Romania.

This study also has some practical implications. Based on previous research [e.g., 11], we know that various organizational strategies, such as decision-making discretion, minimizing incivility, or providing performance, cultivate thriving at work, which in turn promotes the development of individuals but also contributes to positive health. Therefore, the NAP should create an environment to encourage superiors to provide feedback on employee performance, reduce incivility, and thus increase thriving among correctional officers.

The results also showed that thriving is related to PsyCap. Being a personal resource, it is malleable and can be improved through specific training and interventions, helping correctional officers better adapt to working conditions. According to a meta-analysis by LupÈ™a et al. [45], this training can last from one hour to four weeks and consists of exercises to increase employees' psychological resources. These resources protect employees against burnout, which, as we have seen, affects job satisfaction and performance.”

R2: Finally, it would be advisable to explain how, in the profession in which the sample works, and in the organizational context in which they work, the results of this article can serve to improve the quality of work life, reduce burnout and increase organizational performance.

RESPONSE: Finally, we have added a final paragraph in the Theoretical and Practical implications section to explain from a practical point of view how the research results help the correctional officers and indicate how promoting work contexts characterized by thriving can turn organizations into healthier environments (p.10): As practical implications, Romanian correctional officers who experiment thriving at work (i.e., feel energetic at work and accumulate knowledge in the workplace) will develop positive emotions related to the confidence that they can complete a task or have an attitude that positively evaluates everything around them, or have a belief that they will achieve their goals and that they can recover after facing a problem. The correctional officer characterized by these positive states will experience less exhaustion and cynicism in the future, which will make him more satisfied at work and report a higher performance. Thus, thriving climate matters for correctional officers because it enhances employees' health and positive adaptation to work stressors, creating a healthier organizational environment.

R2: I congratulate the authors for their work.

Warm regards,

RESPONSE: Thank you very much once again for the feedback and suggestions provided! We hope that our additions are clear and welcome.